# Learning Non-Gradient Diffusion Systems via Moment-Evolution and Energetic Variational Approaches

## Abstract

This paper proposes a data-driven learning framework for identifying governing laws of generalized diffusions with non-gradient components. By combining energy dissipation laws with a physically consistent penalty and first-moment evolution, we design a two-stage method to recover the pseudo-potential and rotation in the pointwise orthogonal decomposition of a class of non-gradient drifts in generalized diffusions. Our two-stage method is applied to complex generalized diffusions including dissipation-rotation dynamics, rough pseudo-potentials and noisy data. Representative numerical experiments demonstrate the effectiveness of our approach for learning physical laws in non-gradient generalized diffusions.

## 1 Introduction

Over the past several decades, numerous studies have been devoted to nonlinear stochastic dynamics, particularly with regard to entropy production Esposito (2012); Crooks (1999); Qian et al. (1991), fluctuation Kurchan (1998); Marconi et al. (2008); Esposito & den Broeck (2010), power dissipation Ge & Qian (2010), circulation Qian & Qian (1982); Qian & Wang (1999), and non-equilibrium steady states Dorfman (1999); Jiang & Jiang (2004). Through sustained investigation, researchers have gradually come to recognize that the violation of detailed balance plays a key role in revealing the aforementioned fundamental phenomena. Non-detailed balance as an inherent property of diffusion processes underlies the emergence of Hamiltonian conservative dynamics and entropy production. These mechanisms are central to the behavior of living systems in biology and chemistry, as illustrated by models such as the human stem-cell network Li & Wang (2013) and cell-fate decision dynamics Chen et al. (2023). They also appear in broader physical systems, including ocean-current transport models Petrović et al. (2025).

The incorporation of non-gradient structures into SDEs provides a prototypical framework for modeling non-equilibrium dynamics without detailed balance. Such systems are mathematically characterized by irreversible Markov processes. The study of stochastic dynamics without detailed balance often relies on decomposing the generator of the Markov process into symmetric and anti-symmetric parts Qian et al. (1991); Jiang & Jiang (2004); Qian (2013); Qian & Wang (1999). From a partial differential equation (PDE) perspective, especially through the Fokker-Planck (FP) equation, Qian (2013) shows that general diffusions without detailed balance can be systematically decomposed into a reversible stochastic process with detailed balance and a canonical conservative dynamics. According to the Helmholtz decomposition, a vector field can typically be decomposed into two $L^2$-orthogonal components: a gradient component and a rotational component. When these two components are pointwise orthogonal, the gradient part governs the behavior of (non)equilibrium steady states, while the rotational part influences the dynamics through which the system approaches these steady states. This highlights that both the gradient and rotational components play essential roles in the system's evolution. In particular, the rotational component is key to understanding the overall dynamical behavior. Such a decomposition, in the context of the FP equation, is also closely related to large deviation theory. We refer readers to results on the Wentzel-Kramers–Brillouin (WKB) ansatz and the Hamilton-Jacobi equation Graham & Haken (1971); Graham (1973).

One active area of research in the study of SDEs is the discovery of underlying physical laws from observational data. The main idea is to train neural networks or models based on parametric or nonparametric techniques by minimizing a suitable loss function. This loss is often constructed using, for example, probabilistic methods Dietrich et al. (2023); Chen & Xiu (2024); Churchill

& Xiu (2023); Liu et al. (2024); Yang et al. (2020), strong- Brunton et al. (2016); Raissi et al. (2019); Chen et al. (2021a) and weak-forms Zang et al. (2020); Messenger et al. (2024); Ryck et al. (2022); Messenger & Bortz (2022; 2021); Gao et al. (2022) of differential equations or variational structures Huang et al. (2024b); Gruber et al. (2023); Lee et al. (2021); Gruber et al. (2025); Yu et al. (2021); Chen et al. (2024); Huang et al. (2022); Zhang et al. (2022); Kharazmi et al. (2021); Gao et al. (2024); Lu et al. (2024). In Lu et al. (2024), the authors propose a loss function based on a variational structure derived from energy laws, and the proposed algorithm performs effectively in learning potential-driven dissipative systems. However, their framework does not address diffusion processes that include rotational components.

Recent efforts have explored data-driven methods for learning quasi-potentials associated with non-gradient stochastic dynamics. As shown in Lin et al. (2022); Li et al. (2022); Grigorio & Alqahtani (2024), the learning of quasi-potential in the pointwise orthogonal decomposition of drift within FP equations has rich applications in engineering, biology, etc. The computational method proposed in Lin et al. (2022); Li et al. (2022); Grigorio & Alqahtani (2024) for learning the quasi-potential is to minimize the loss function generated by the governing differential equations. In large deviation theory, the computation of the quasi-potential also involves non-gradient structures. However, such analyses primarily focus on small perturbations around (meta)steady states, rather than the global behavior of the entire system. This is subtly but fundamentally different from the goal of the present work, which is to investigate the global dynamics of generalized diffusions with non-gradient structures. Extending our methodology to compute quasi-potentials remains an interesting and promising direction for future research.

In this paper, we focus on the learning of governing laws in generalized diffusions with non-gradient structures. Motivated by the learning framework established in Lu et al. (2024), without relying on governing FP equations, we learn the drift and pseudo-potential by combining first-moment and energy laws. *Here the pseudo-potential is defined as the rate function of the stationary distribution satisfying the WKB ansatz form.* Concerning a class of drift terms with rotation components satisfying pointwise orthogonal decomposition in stochastic processes, we perform numerical experiments in dimension two. Our two-stage learning framework is an extension of that shown in Lu et al. (2024) and has contributed to the following aspects

- We develop a novel two-stage framework consisting of first-moment evolution and an energy dissipation law for learning the decomposition of a general class of drifts with rotation components in nonlinear stochastic dynamics. One of the most significant advantages is that the loss functions are formulated as integral forms, which require low regularity of drifts and data observation.

- In the loss function based on the energy dissipation law, we propose a physically consistent penalty, derived via dimensional analysis, that aims to orthogonally decompose the non-gradient drift into pseudo-potential and rotational components pointwise.

- We investigate the effectiveness of our algorithm for learning physical laws including drifts, pseudo-potentials and rotations over various representative generalized diffusions with different hyperparameters.

- We showcase the effectiveness of learning rough pseudo-potentials and robustness to noisy data observation.

The objective of this paper is to propose an alternative (weak-form) learning approach that complements existing PDE-based (strong-form) methods. Weak-form loss functions generally offer greater robustness against noisy observations compared to their strong-form counterparts. On the other hand, strong-form formulations are better suited to capturing local information. Therefore, integrating these two approaches has the potential to create a more effective loss function for learning physical laws. These possibilities will be explored in future research.

This paper is organized as follows. In Section 2, we introduce the mathematical formulation underlying our learning algorithm. Section 3 presents the learning framework of our two-stage method: in Stage 1, we parameterize the loss function using the first-moment evolution; in Stage 2, we adopt the learning strategy of Lu et al. (2024) to minimize the loss function based on the energy law and recover the potential. Section 4 provides several numerical examples to illustrate the effectiveness and robustness of our approach. It also includes ablation studies comparing (a) the proposed two-stage method with direct approaches, and (b) the proposed weighted penalty with the standard $L_2$

penalty. Section 5 discusses related works, including strong-form-based methods and variational- or weak-form-based approaches. Finally, Section 6 summarizes our main findings and discusses several open problems for future research.

## 2 FORMULATION

We consider a dynamical system with some white noise perturbation. The corresponding evolution of the state variable $\mathbf{X}_t$ is described by the following SDE:

$$d\mathbf{X}_t = \mathbf{b}(\mathbf{X}_t)dt + \sigma(\mathbf{X}_t)d\mathbf{W}_t, \ \mathbf{X}_t = (x_1^t, \cdots, x_d^t) \in \mathbb{R}^d, \tag{1}$$

where time $t \geq 0$, drift $\mathbf{b} : \mathbb{R}^d \to \mathbb{R}^d$ is a continuously differentiable vector field, $\mathbf{W}_t$ is a standard $d$-dimensional Brownian motion and the noise intensity $\sigma : \mathbb{R}^d \to \mathbb{R}$ is a scalar function. We restrict attention to a scalar noise coefficient $\sigma$ in order to exclude correlated noise effects. This allows us to focus solely on learning the drift decomposition in non-gradient dynamical systems. For related work on stochastic dynamics driven by correlated noise and their learning via neural network–based approaches, we refer the reader to Guo et al. (2025).

Applying the classical Itô integral formula on (1), one has the associated FP equation is

$$\partial_t f + \nabla \cdot (\mathbf{b}f) = \frac{1}{2}\Delta(\sigma^2 f), \tag{2}$$

where $f(\mathbf{x}, t)$ is the probability density function of $\mathbf{X}_t$ at time $t$. Here we assume $f_0 := f(\mathbf{x}, 0)$ is the density of initial state $\mathbf{X}_0$. Next, we derive the loss function from the underlying energy law; the detailed derivation is given in Appendix A.

Multiplying (2) by $x_i$, $i = 1, \cdots, d$, we use integration by parts to get (2) admits the following first-moment evolution:

$$\frac{d}{dt} \int_{\mathbb{R}^d} f x_i d\mathbf{x} = \int_{\mathbb{R}^d} \mathbf{b} \cdot \mathbf{e}^{(i)} f d\mathbf{x} - \frac{1}{2} \int_{\mathbb{R}^d} \partial_{x_i}(\sigma^2 f) d\mathbf{x} = \int_{\mathbb{R}^d} \mathbf{b} \cdot \mathbf{e}^{(i)} f d\mathbf{x}. \tag{3}$$

Since our goal is to extract both pseudo-potential and rotation component in generalized diffusions, the information provided by (3) alone is insufficient. Next, we formulate the energy laws satisfied by the FP equations without detailed balance.

We consider that the drift $\mathbf{b}$ has the following form:

$$\mathbf{b} = -\frac{1}{2}\sigma^2 \nabla \psi + \frac{1}{2}\sigma^2 \mathbf{R}, \tag{4}$$

where $\psi$ is the potential function, $\sigma$ is the noise intensity and $\mathbf{R}$ is the rotation part. Adopting the following free energy possessed by potential FP equations Lu et al. (2024), we have

$$\mathcal{F}(t) = \int_{\mathbb{R}^d} \left[ f \ln\left(\frac{1}{2}\sigma^2 f\right) + \psi f \right] d\mathbf{x}, \tag{5}$$

and its evolution satisfies

$$\frac{d\mathcal{F}}{dt} = -\int_{\mathbb{R}^d} \frac{2f}{\sigma^2} |\mathbf{u}|^2 \, d\mathbf{x} + \frac{1}{2} \int_{\mathbb{R}^d} \left[ \mathbf{R} \cdot \nabla(\sigma^2 f) + \sigma^2 f \mathbf{R} \cdot \nabla \psi \right] d\mathbf{x}, \tag{6}$$

where $\mathbf{u} := -\left[\frac{\sigma^2}{2}\nabla \ln(\sigma^2 f) + \frac{\sigma^2}{2}\nabla \psi\right]$ denotes the average velocity. Moreover, similarly as the definition of quasi-potential shown in Lin et al. (2022), we suppose $\psi$ and $\mathbf{R}$ satisfy

$$\nabla \psi \cdot \mathbf{R} = 0, \tag{7}$$

where $\mathbf{R}$ is the rotational component. It follows from (6) that

$$\frac{d\mathcal{F}}{dt} = -\int_{\mathbb{R}^d} \frac{2f}{\sigma^2} \left| \frac{\sigma^2}{2}\nabla \ln(\sigma^2 f) + \frac{\sigma^2}{2}\nabla \psi \right|^2 \, d\mathbf{x} \leq 0, \tag{8}$$

which is dissipative in time $t$. In other words, with the pointwise orthogonality condition $\nabla \psi \cdot \mathbf{R} = 0$, we can minimize the (8) in a weak form to learn the pseudo-potential and the rotation part. More

precisely, since the dissipation law (8) holds for a given density function $f$ satisfying FP equation (2), we determine the pseudo-potential $\psi$ by minimizing

$$\left| \frac{d\mathcal{F}}{dt} + \int_{\mathbb{R}^d} \frac{2f}{\sigma^2} |\mathbf{u}|^2 \, d\mathbf{x} \right|^2, \tag{9}$$

taking advantage of the fact that $\psi$ is time-independent. Here, $f$ is treated as known data, $\sigma$ is given, and $\mathcal{F}$ is defined in (5). It is worth noting that $\psi$ is the sole unknown in (9). Subsequently, if $\mathbf{b}$ is learned from (3), the rotational component $\mathbf{R}$ can then be determined via the decomposition (4).

To determine the form of penalty $\mathcal{P}$ associated with the orthogonality constraint in the learning process, we apply the dimensional analysis outlined in Appendix B.2, which yields

$$\mathcal{P} := \left[ \int_{\mathbb{R}^d} \sigma^2 f |\nabla \psi \cdot \mathbf{R}| \, d\mathbf{x} \right]^2. \tag{10}$$

## 3 Learning Methods

In the context of FP equations with non-gradient structures, our study introduces a two-stage approach for learning the underlying physical laws. Suppose a generalized form of fluctuation-dissipation relation (4) holds and continuous data observation is available, our objective is to identify the pseudo-potential and the rotation component. We shall propose the learning framework and investigate the effect of data property on the learning results.

We take advantage of the approach proposed in Lu et al. (2024) and shall utilize the energy law (8) to construct the loss function. Whereas, due to the presence of rotation component, we are not able to learn the pseudo-potential by only leveraging the energy law. To this end, we first construct the loss function based on first-moment evolution (3) and learn the general drift $\mathbf{b}$, then formulate the second loss function by the energy law and investigate the pseudo-potential.

Assuming the continuous data observation is available, we implement our approach by learning the pseudo-potential $\psi$ with known noise intensity $\sigma^2$. Here we consider the unknown pseudo-potential function $\psi(x)$ is approximated by a neural network $\psi_{NN}(x; \theta)$. Before introducing our two-stage method, we discuss the choice of training data in detail as follows.

Let $f_j(\mathbf{x}, t)$ be the solution to the FP equation evolving from Gaussian-type initial data $(f_0)_j(\mathbf{x})$ with mean $\boldsymbol{\mu}_j^0$ and variance $\sigma_0^2$, for $j = 1, \ldots, M$, where $M$ is the number of datasets. Let $\Delta x_k$, $k = 1, \ldots, d$, be the uniform spatial mesh size and $\Delta t$ be the time step used in the FP solver. Our training data consist of the observation dataset $\{f_j(\mathbf{x}_i, t_1), f_j(\mathbf{x}_i, t_2), f_j(\mathbf{x}_i, T_1), f_j(\mathbf{x}_i, T_2)\}_{i,j}^{N,M}$, where $t_2 = t_1 + \delta t$ and $T_2 = T_1 + \delta t$. Here, $\delta t = m\Delta t$ is a prescribed observation time step size, $t_1 \ll 1$ is the short (transient) timescale for the first stage, and $T_1 > t_1$ corresponds to a longer, intermediate timescale for the second stage. The spatial points $\{\mathbf{x}_i\}$ form a uniform mesh of size $\delta x_k = n\Delta x_k$ (observation spatial grid size). Now, we outline our two-stage approach as follows.

### 3.1 Stage 1: Moment Estimate

Define $\boldsymbol{\mu}_j = ((\mu_j)_1, \ldots, (\mu_j)_d)$ as the centroid of density function $f$ to (2), which can be approximated by $(\mu_j)_k(t) = |\delta\mathbf{x}| \sum_{i=1}^N (x_i)_k f_j(\mathbf{x}_i, t)$, $j = 1, \ldots, M$, $k = 1, \ldots, d$, where the spatial variable $\mathbf{x}_i$ satisfying $\mathbf{x}_i = ((x_i)_1, \ldots, (x_i)_d)$ and $\delta\mathbf{x} = (\delta x_1, \ldots, \delta x_d)$. Define drift as $\mathbf{b} = (b_1, \ldots, b_d)$ and

$$\theta_{\mathbf{b}}^* = \underset{\theta}{\arg\min} \, L_{\mathbf{b}}^{\text{dyn}}(\theta), \tag{11}$$

where $L_{\mathbf{b}}^{\text{dyn}}(\theta) = \sum_{k=1}^d L_{b_k}^{\text{dyn}}(\theta)$ and

$$L_{b_k}^{\text{dyn}}(\theta) = \sum_{j=1}^M \left\| \frac{(\mu_j)_k(t_2) - (\mu_j)_k(t_1)}{t_2 - t_1} - |\delta\mathbf{x}| \sum_{i=1}^N (b_k)_{\text{NN}}(\mathbf{x}_i; \theta) f_j(\mathbf{x}_i, t_1) \right\|^2, \quad k = 1, \ldots, d. \tag{12}$$

Here $L_{b_k}^{\text{dyn}}$ is obtained by approximating (3) in terms of the Riemann sum. We remark that (12) is convex in $(b_k)_{\text{NN}}$, which implies (11) is unique and the convex problem performs well numerically.

After learning $(b_k)_{\text{NN}}$ by (11), we next learn the pseudo-potential $\psi$ based on energy laws, which is shown as follows.

## 3.2 Stage 2: Energy Dissipation Laws

We use the Riemann sum to approximate free energy (5) and obtain $\mathcal{F}_j(t; \theta) = |\delta\mathbf{x}| \sum_{i=1}^{N} \left[ f_j(\mathbf{x}_i, t) \ln\left(\frac{1}{2}\sigma^2(\mathbf{x}_i) f_j(\mathbf{x}_i, t)\right) + \psi_{\text{NN}}(\mathbf{x}_i; \theta) f_j(\mathbf{x}_i, t) \right]$. Based on this, we discretize (6) to get the following loss function $L_\psi^{\text{dyn}}(\theta)$

$$L_\psi^{\text{dyn}}(\theta) = \sum_{j=1}^{M} \left\| \frac{\mathcal{F}_j(T_2; \theta) - \mathcal{F}_j(T_1; \theta)}{T_2 - T_1} + |\delta\mathbf{x}| \sum_{i=1}^{N} \frac{1}{2}\sigma^2(\mathbf{x}_i) f_j(\mathbf{x}_i, T_1) \left| \tilde{\nabla} \ln(\sigma^2(\mathbf{x}_i) f_j(\mathbf{x}_i, T_1)) + \nabla\psi_{\text{NN}}(\mathbf{x}_i; \theta) \right|^2 \right\|^2, \tag{13}$$

where $\tilde{\nabla}$ is the numerical gradient computed using data. Due to the orthogonality condition shown in (7), we define the penalty as

$$L_\psi^{\text{orth}}(\theta; \mathbf{b}^*) = \sum_{j=1}^{M} \left\| |\delta\mathbf{x}| \sum_{i=1}^{N} \sigma^2(\mathbf{x}_i) f_j(\mathbf{x}_i, T_1) \left| \nabla\psi_{\text{NN}}(\mathbf{x}_i; \theta) \cdot \left( \frac{2}{\sigma^2(\mathbf{x}_i)} \mathbf{b}^*(\mathbf{x}_i) + \nabla\psi_{\text{NN}}(\mathbf{x}_i; \theta) \right) \right| \right\|^2, \tag{14}$$

where $\mathbf{b}^*(\mathbf{x}_i) := \mathbf{b}_{\text{NN}}(\mathbf{x}_i; \theta_{\mathbf{b}}^*)$ in which $\theta_{\mathbf{b}}^*$ is given by (11) in Stage 1. Combining (13) and (10), we formulate the following optimization problem for learning pseudo-potential $\psi$:

$$\theta_\psi^* = \operatorname*{argmin}_\theta \{L_\psi^{\text{dyn}}(\theta) + \lambda L_\psi^{\text{orth}}(\theta; \mathbf{b}^*)\} := \operatorname*{argmin}_\theta \mathcal{L}_\lambda(\theta), \text{where } \lambda \text{ is a multiplier.} \tag{15}$$

We summarize the learning framework shown in Subsections 3.1 and 3.2 as the following diagram:

---

**Algorithm 1** Learning non-gradient diffusions using the two-stage method

- Given probability density functions of four time steps $\{(f_j(\mathbf{x}_i, t_1), f_j(\mathbf{x}_i, t_2), f_j(\mathbf{x}_i, T_1), f_j(\mathbf{x}_i, T_2))\}_{i,j=1}^{N,M}$ for training.

- Stage 1: Learn the general drift $\mathbf{b}$ by optimizing the loss function (12) and find the "best" parameters of the neural networks to reconstruct $\mathbf{b}_{\text{NN}}$.

- Stage 2: Learn the pseudo-potential $\psi$ by optimizing the loss function $\mathcal{L}_\lambda$ given in (15) and find the "best" parameters of the neural networks to reconstruct $\psi_{\text{NN}}$.

---

In addition, we present the following flowchart to provide a depiction of our learning framework:

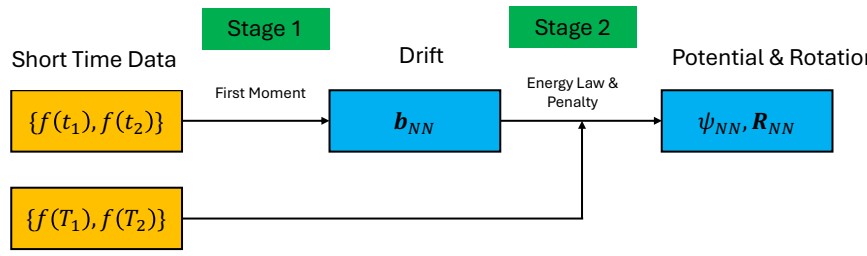

Figure 1: The flowchart of the two-stage method.

We remark that in our two-stage learning framework for nonlinear stochastic dynamics, we consider two temporal steps: in Stage 1, the short time is chosen for training to capture the drift $\mathbf{b}$, while in Stage 2, we choose the intermediate time to learn the pseudo-potential $\psi$. The reason is that, in Stage 1, short-time data remain distinct enough to be a family of effective test functions, allowing for a better identification of the drift compared to long-term data. In addition, the intermediate temporal data in Stage 2 can reduce the non-convexity of (13) significantly based on the free energy since as $t$ large enough, (13) has a unique solution $\psi_{\text{NN}} = -\ln(\sigma^2 f)$.

## 4 Numerical Examples

In this section, we show several representative numerical examples and focus on two- and three-dimensional cases, i.e. $d = 2, 3$. Noting that the equation (2) is posed on the whole space $\mathbb{R}^2$, we solve the FP equation on a sufficiently large computational domain, e.g. $[-4, 4]^2$ for the 2D cases, to achieve an accurate approximation. To generate continuous data observations, we simulate $M$ different initial distributions of $\mathcal{N}(\boldsymbol{\mu}^0, \sigma_0^2 I)$, where the $M$ mean vectors $\boldsymbol{\mu}^0$ are uniformly spaced in a smaller domain, e.g.$[-2, 2]^2$ in the 2D cases, $\sigma_0^2 = 0.01$, and $I$ is the identity matrix. We remark that in stage 2, while learning pseudo-potentials based on energy laws, there are singularities caused by $\ln f$ and $|\nabla f|^2 / f$ when the density $f$ vanishes in (13). To tackle this numerical issue, we approximate $f$ by using $\tilde{f} = \max(f, 5 \times 10^{-4})$.

Note that the pseudo-potential $\psi$ is uniquely determined up to an additive constant. To compare the neural network approximation $\psi_{\text{NN}}$ with the ground truth $\psi$, we shift $\psi_{\text{NN}}$ so that the average of $\psi_{\text{NN}}$ is the same as that of ground truth $\psi$: $\psi_{\text{NN}}(\mathbf{x}) \to \psi_{\text{NN}}(\mathbf{x}) + \frac{1}{N} \sum_{i=1}^{N} (\psi(\mathbf{x}_i) - \psi_{\text{NN}}(\mathbf{x}_i))$.

For training in both stages, we use a four-layer fully connected neural network with two 50-node hidden layers and **Tanh()** activation. Training uses the **Adam** optimizer (learning rate $10^{-4}$), batch size 5, for $10,000$ epochs.

We evaluate the performance of each step of our method using the following relative root mean square errors (rRMSE): $\text{rRMSE}_{\mathbf{b}} = \sqrt{\sum_{i=1}^{N} |\mathbf{b}(\mathbf{x}_i) - \mathbf{b}_{\text{NN}}(\mathbf{x}_i; \theta_{\mathbf{b}}^*)|^2} / \sqrt{\sum_{i=1}^{N} |\mathbf{b}(\mathbf{x}_i)|^2}$, $\text{rRMSE}_{\psi} = \sqrt{\sum_{i=1}^{N} (\psi(\mathbf{x}_i) - \psi_{\text{NN}}(\mathbf{x}_i; \theta_{\psi}^*))^2} / \sqrt{\sum_{i=1}^{N} (\psi(\mathbf{x}_i))^2}$, $\text{rRMSE}_{\mathbf{R}} = \sqrt{\sum_{i=1}^{N} |\mathbf{R}(\mathbf{x}_i) - \mathbf{R}_{\text{NN}}(\mathbf{x}_i)|^2} / \sqrt{\sum_{i=1}^{N} |\mathbf{R}(\mathbf{x}_i)|^2}$, where $\mathbf{R}_{\text{NN}} := \frac{2}{\sigma^2} \mathbf{b}_{\text{NN}} + \nabla \psi_{\text{NN}}$.

### 4.1 An illustrative example

To demonstrate the effectiveness of our two-stage method, we begin with a basic two-dimensional example in which the potential is a double-well, and the rotational component is the orthogonal complement of the potential's gradient (i.e., the *canonical* form).

**Double-well Potential & Canonical Rotation.** Consider the potential function $\psi(x, y) = \frac{1}{4}(x^2 - 1)^2 + \frac{1}{2}y^2$, the rotation $\mathbf{R}(x, y) = \nabla \psi^\perp = \left[ y, -(x^3 - x) \right]^T$, and the noise intensity $\sigma^2(x, y) = 1/(1 + x^2 + y^2)$. Then, the ground truth drift is given by $\mathbf{b}(x, y) = \frac{1}{2(1+x^2+y^2)} \left[ -(x^3 - x) + y, -y - (x^3 - x) \right]^T$.

The training data $\{f_j(x_i, y_i, t_1), f_j(x_i, y_i, t_2), f_j(x_i, y_i, T_1), f_j(x_i, y_i, T_2)\}_{i,j=1}^{N,M}$ are generated by solving the FP equation using a spatial grid size of $\Delta x = \Delta y = 0.1$ and a time step size of $\Delta t = 0.0001$. For the baseline experiment, we simulate 40 different initial distributions and select snapshots at $t_1 = 0.015$, $t_2 = 0.016$, $T_1 = 2$ and $T_2 = 2.001$ as our training data. We set the weight $\lambda = 10$ for the orthogonality penalty in (15). Applying the two-stage method yields a learned drift vector field and a learned pseudo-potential, correspondingly a learned rotation field (Figures 3,4,5).

Table 1 summarizes a series of experiments with varying hyperparameters, including the sample size $M$, observation grid size $\delta x$ and $\delta y$, observation time step size $\delta t$, timescales $t_1$ and $T_1$ for the first and second stages respectively, and the weight $\lambda$ for the orthogonality penalty.

As shown in Table 1, $\lambda = 10$ leads to a relatively small error in learning the pseudo-potential and rotation term compared to $\lambda = 1$ and $\lambda = 100$. Furthermore, learning drifts and pseudo-potentials are robust to the increment of $\delta t$ and $t_1$. Whereas, an increase in $\delta x$ adversely affects the ability to learn the drift. It should be noted, however, that $\delta x = 0.1$ is already a relatively coarse grid size. Similarly, decreasing $T_1$ leads to a worse learning of pseudo-potentials, as it results in a more severely non-convex loss function, as discussed in Lu et al. (2024). In particular, Experiment 2 shown in Table 1 implies that the sparseness of data observation (the number $M$ of the initial distributions is small) significantly reduces the accuracy of learning.

Table 1: Experiments of the basic example using different hyperparameters, along with their corresponding relative root mean square errors.

| Experiment | $M$ | $\delta x$ $(\delta y)$ | $\delta t$ | $t_1$ | $T_1$ | $\lambda$ | rRMSE$_\mathbf{b}$ | rRMSE$_\psi$ | rRMSE$_\mathbf{R}$ |
|---|---|---|---|---|---|---|---|---|---|
| 1 (baseline) | 40 | 0.1 | 0.001 | 0.015 | 2 | 10 | 2.266e-02 | 1.929e-02 | 3.495e-02 |
| 2 | 20 | 0.1 | 0.001 | 0.015 | 2 | 10 | 3.702e-01 | 3.007e-01 | 6.286e-01 |
| 3 | 40 | 0.2 | 0.001 | 0.015 | 2 | 10 | 4.610e-02 | 1.968e-02 | 5.837e-02 |
| 4 | 40 | 0.1 | 0.01 | 0.015 | 2 | 10 | 1.784e-02 | 1.605e-02 | 3.101e-02 |
| 5 | 40 | 0.1 | 0.001 | 0.5 | 2 | 10 | 2.633e-02 | 1.345e-02 | 3.435e-02 |
| 6 | 40 | 0.1 | 0.001 | 0.015 | 1 | 10 | 2.230e-02 | 2.552e-02 | 4.661e-02 |
| 7 | 40 | 0.1 | 0.001 | 0.015 | 2 | 1 | 1.551e-02 | 6.259e-02 | 8.778e-02 |
| 8 | 40 | 0.1 | 0.001 | 0.015 | 2 | 100 | 1.838e-02 | 5.807e-01 | 9.840e-01 |

## 4.2 Further analysis

We now present several examples to test the robustness of our two-stage method. The first example uses the same double-well potential as in Section 4.1, but incorporates a non-canonical rotation. The second example involves a quadruple-well potential combined with the canonical rotation, along with an oscillatory noise intensity. The third example examines a rough double-well potential. The fourth example revisits the original double-well potential and canonical rotation from Section 4.1, but with training data contaminated by Gaussian noise. In the fifth example, we consider a three-dimensional problem with a double-well potential and a canonical rotation to show the effectiveness of our two-stage method in higher-dimensional problems. In the sixth example, we examine the impact of different choices of training-data time stamps, which quantifies how close the training data are to the steady state(s), on obtaining reasonable learning results. In the seventh example, we validate our method on particle data, which are more readily available in real-world applications. Finally, we compare our two-stage method with a PDE-based approach on a simple example to demonstrate the necessity of the two-stage framework for learning non-gradient systems.

**Double-well Potential & Non-canonical Rotation.** This example demonstrates that the two-stage method remains effective for rotational fields $\mathbf{R}$ that are not of the form $\mathbf{R} = \nabla\psi^\perp$.

Let $\psi$, $\mathbf{R}$, and $\sigma^2$ be the same as in the example of Section 4.1. We define $\overline{\psi} = \psi$, $\overline{\mathbf{R}} = \frac{\psi}{4}\mathbf{R}$, where the factor $\frac{1}{4}$ is introduced to avoid numerical overflow near the boundary. Note that $\nabla\overline{\psi}\cdot\overline{\mathbf{R}} = \frac{\psi}{4}(\nabla\psi\cdot\mathbf{R}) = 0$ and $\nabla\cdot\overline{\mathbf{R}} = \frac{\nabla\psi}{4}\cdot\mathbf{R} + \frac{\psi}{4}\nabla\cdot\mathbf{R} = 0$ since $\nabla\psi\cdot\mathbf{R} = 0$ and $\nabla\cdot\mathbf{R} = 0$.

In other words, without ambiguity in notation, we consider the potential function $\psi(x,y) = \frac{1}{4}(x^2 - 1)^2 + \frac{1}{2}y^2$, the rotation $\mathbf{R}(x,y) = \left(\frac{1}{16}(x^2 - 1)^2 + \frac{1}{8}y^2\right)\left[y, -(x^3 - x)\right]^T$. Thus, the ground truth drift is given by $\mathbf{b}(x,y) = \frac{1}{2(1+x^2+y^2)}\left[-(x^3 - x) + \left(\frac{1}{16}(x^2 - 1)^2 + \frac{1}{8}y^2\right)y, -y - \left(\frac{1}{16}(x^2 - 1)^2 + \frac{1}{8}y^2\right)(x^3 - x)\right]^T$.

We first adopt the same hyperparameter setting as in the baseline of the example in Section 4.1, with a sample size of $M = 40$. We observe that while the learned drift $\mathbf{b}_{\text{NN}}$ and potential $\psi_{\text{NN}}$ closely match the ground truth (rRMSE$_\mathbf{b}$ = $1.232 \times 10^{-1}$ and rRMSE$_\psi$ = $6.287 \times 10^{-2}$), the reconstructed rotational field $\mathbf{R}_{\text{NN}}$ deviates more significantly from the true $\mathbf{R}$ (rRMSE$_\mathbf{R}$ = $5.290 \times 10^{-1}$). To accurately capture the complex structure of $\mathbf{R}$, a larger sample size is needed. We increase $M$ to 80, and the resulting learned field $\mathbf{R}_{\text{NN}}$ is shown in Figure 6. The relative root mean square errors are rRMSE$_\mathbf{b}$ = $3.640 \times 10^{-2}$, rRMSE$_\psi$ = $4.337 \times 10^{-2}$, and rRMSE$_\mathbf{R}$ = $1.692 \times 10^{-1}$.

**Quadruple-well Potential & Canonical Rotation.** In this example, we consider a symmetric quadruple-well potential, $\psi(x,y) = \frac{1}{8}(x^2 - 1)^2 + \frac{1}{8}(y^2 - 1)^2$, paired with the rotational field $\mathbf{R}(x,y) = \frac{1}{2}\left[y^3 - y, -(x^3 - x)\right]^T$, and an oscillatory noise intensity $\sigma^2(x,y) = 1 + \frac{1}{2}\cos((x + \frac{1}{2})^2 + y^2)$. The corresponding ground truth drift field is given by $\mathbf{b}(x,y) = \left(\frac{1}{4} + \frac{1}{8}\cos((x + \frac{1}{2})^2 + y^2)\right)\left[-(x^3 - x) + (y^3 - y), -(y^3 - y) - (x^3 - x)\right]^T$.

We use the same hyperparameter settings as in the baseline of example in Section 4.1 with $M = 80$. The learned pseudo-potential is shown in Figure 7. The relative root mean square errors are rRMSE$_\mathbf{b}$ = $1.441 \times 10^{-1}$, rRMSE$_\psi$ = $1.513 \times 10^{-1}$, and rRMSE$_\mathbf{R}$ = $2.265 \times 10^{-2}$.

**Rough Double-well Potential & Canonical Rotation.** In this example, we consider the potential function $\psi(\mathbf{x}) = \frac{1}{4}(x^2 - 1)^2 + \frac{1}{2}y^2 + \varepsilon^4 \sin(\frac{2\pi x}{\varepsilon}) \sin(\frac{2\pi y}{\varepsilon})$ with a parameter $\varepsilon$. The parameter $\varepsilon$ controls the oscillatory behavior of the potential function and its derivatives — smaller values of $\varepsilon$ lead to stronger oscillations. Moreover, the oscillations become more prominent in higher-order derivatives compared to lower-order ones. In other words, under a given resolution, higher-order derivatives are more difficult to estimate accurately compared to lower-order ones. For the reader's convenience, we also list the gradient of the potential function $\nabla\psi(x, y) = \left[ x + 2\pi\varepsilon^3 \cos(\frac{2\pi x}{\varepsilon}) \sin(\frac{2\pi y}{\varepsilon}), y + 2\pi\varepsilon^3 \sin(\frac{2\pi x}{\varepsilon}) \cos(\frac{2\pi y}{\varepsilon}) \right]^T$, the rotation term $\mathbf{R}(x, y) = \left[ -y - 2\pi\varepsilon^3 \sin(\frac{2\pi x}{\varepsilon}) \cos(\frac{2\pi y}{\varepsilon}), x + 2\pi\varepsilon^3 \cos(\frac{2\pi x}{\varepsilon}) \sin(\frac{2\pi y}{\varepsilon}) \right]^T$, and the noise intensity $\sigma^2 = 2$ here.

In this example, we choose $\varepsilon = 0.4$, the corresponding first and second partial derivatives of the potential function are shown in Figure 2. We use the same hyperparameter setting as in the example in Section 4.1, except we use a lower resolution $\delta x = \delta y = 0.2$ for training. The learned pseudo-potential is shown in Figure 8. The relative root mean square errors are $\text{rRMSE}_{\mathbf{b}} = 5.670 \times 10^{-2}$, $\text{rRMSE}_{\psi} = 3.539 \times 10^{-2}$, and $\text{rRMSE}_{\mathbf{R}} = 9.690 \times 10^{-2}$.

**The Impacts of Noisy Data.** This example aims to test the robustness of our proposed two-stage method for noisy training data. For comparison and convenience, we follow the same potential function and rotation term as in the example in Section 4.1.

The training data $\{f_j(x_i, y_i, t_1), f_j(x_i, y_i, t_2), f_j(x_i, y_i, T_1), f_j(x_i, y_i, T_2)\}_{i,j=1}^{N,M}$ are generated by solving the FP equation as in the example in Section 4.1. To construct noisy training data, the numerical solution $f$ is convoluted with a Gaussian distribution $\mathcal{N}(\mathbf{0}, \gamma I)$, where $I$ is the identity matrix and the parameter $\gamma$ is referred to as the noise level.

We use the same hyperparameters as in the first row of Table 1, except for the noise level. For the reader's convenience, we reproduce the baseline result in the first row of Table 2. As shown in Table 2, although the accuracy of the learned model gradually decreases as the noise intensity increases, our method yields reasonably reliable results, with all errors remaining below 35%, even when the noise level reaches as high as $\gamma = 0.3$. This robustness primarily arises from the fact that our loss function is formulated in an integral form. Moreover, the presence of an underlying variational structure allows us to avoid the challenges associated with approximating higher-order derivatives.

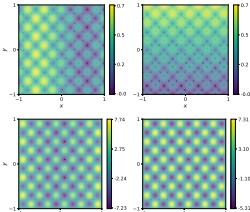

Figure 2: Rough potential. **Upper row:** The first derivatives. **Lower row:** The second derivatives.

Table 2: The Impacts of Noisy Data.

| Experiment | Noise level | rRMSE$_{\mathbf{b}}$ | rRMSE$_{\psi}$ | rRMSE$_{\mathbf{R}}$ |
|---|---|---|---|---|
| 1 | 0.0 | 2.266e-02 | 1.929e-02 | 3.495e-02 |
| 2 | 0.1 | 3.541e-02 | 3.320e-02 | 4.226e-02 |
| 3 | 0.2 | 9.468e-02 | 7.521e-02 | 8.616e-02 |
| 4 | 0.3 | 2.235e-01 | 2.057e-01 | 3.404e-01 |
| 5 | 0.4 | 4.426e-01 | 2.163e-01 | 7.200e-01 |

**Three-dimensional problem.** Consider $\psi(x, y, z) = (1 - x^2)^2 + y^2 + z^2$ and $\mathbf{R}(x, y, z) = \left[ -(y + z), 2(x^3 - x), 2(x^3 - x) \right]^T$. We solve the Fokker-Planck equation on $[-3, 3] \times [-1.5, 1.5]^2$, simulate $M = 80$ initial distributions in $[-2, 2] \times [-1, 1]^2$ for generating training data, and adopt the same hyperparameter setting as in the baseline of the example in Section 4.1 ($\Delta x = \Delta y = \Delta z = 0.1$, $\Delta t = 0.0001$, $t_1 = 0.015$, $t_2 = 0.016$, $T_1 = 2$ and $T_2 = 2.001$). The relative root mean square errors are $\text{rRMSE}_{\mathbf{b}} = 1.568 \times 10^{-1}$, $\text{rRMSE}_{\psi} = 5.991 \times 10^{-2}$, and $\text{rRMSE}_{\mathbf{R}} = 1.596 \times 10^{-1}$. This example demonstrates the robustness of the two-stage method in higher-dimensional problems.

**The impact of different choices of $T_1$.** In our two-stage method, training data at $t_1$ and $T_1$ are used to accommodate the first-moment loss and the EnVarA loss. As noted in Lu et al. (2024), closer second-stage data to the steady state generally improve learning performance. To assess

proximity to steady state, we use the absolute value of free-energy dissipation $\frac{d\mathcal{F}}{dt}$, which vanishes at equilibrium. This example uses the same setup as Section 4.1 but varies $T_1$. As shown in Table 5, choosing $T_1 \geq 1.0$ yields good learning results, consistent with Lu et al. (2024).

**Particle-to-density method.** We evaluate our method on particle data, which are more accessible than probability density data. Consider the same system as in Section 4.1. We solve the SDE with $\Delta t = 0.001$ and simulate $M = 40$ initial distributions with means in $[-1.5, 1.5]^2$ and fixed variance 0.04 for generating training data, where each distribution is approximated by the kernel density estimation method using $10^4$ particles. The training data are chosen at $t_1 = 0.5$, $t_2 = 0.7$ and $T_1 = 2.0$, $T_2 = 2.2$. Therefore, the observational time step size is 0.2. In this particle-to-density setting, as shown in Figures 9, 10, and 11, our method yields reasonable learning results.

**Comparison to a PDE-based method.** To compare our two-stage method with existing PDE-based methods Schaeffer (2017); Lu et al. (2024), which are widely used in science and engineering, we consider a 2D single-well potential $\psi(x, y) = \frac{1}{2}(x^2 + y^2)$. We then minimize a PDE-based loss function aiming to learn the gradient (without rotation) and non-gradient drifts (with a canonical rotation $\mathbf{R}(x, y) = [y, -x]^T$), respectively; the explicit form of this loss is given in Appendix D. The learning results show that the PDE-based method can successfully recover the potential for gradient systems from data, but fails to do so for non-gradient systems. See Figures 12 and 13 in Appendix C.

### 4.3 ABLATION STUDIES

**Direct Methods vs Two-stage Method.** We compare the performance of the two-stage method proposed in Section 3 with two direct methods described in Appendix B.1, which aim to jointly learn the potential component $\psi$ and the rotation component $\mathbf{R}$ from data.

To illustrate, we consider the basic example in Section 4.1 while the noise intensity is set to a constant value $\sigma^2 \equiv 2$. For simulation, we adopt the baseline's hyperparameters in Table 1. The results in Table 3 demonstrate that the two-stage method achieves the most stable and accurate performance in learning all the drift, pseudo-potentials, and rotational components. In particular, although the energy-based direct method performs well in learning the pseudo-potential $\psi$, it exhibits deficient performance in capturing the rotational component. This limitation arises because the long temporal data adversely affects the accurate learning of the drift by solely using an energy law-based loss function. Similarly, first-moment direct method can not achieve the decent approximation of $\psi$ due to the short-time data observation.

**The Impacts of Different Penalties.** We test the effectiveness of our proposed penalty term based on the dimensional analysis mentioned in Appendix B.2. For comparison, we follow the same setting as in the example in Section 4.1 except the penalty term. A comparison between the weighted penalty given by (10) and the standard $L_2$ penalty is provided in Appendix B.2.

As shown in Table 4, the accuracies of the drift term for both penalty types are nearly identical. In fact, they would be exactly the same in the absence of randomness in neural network training, since the first stage of the method does not involve the penalty term. However, the standard penalty fails to properly decompose the gradient and rotational components, as its loss function generally lacks the ability to effectively control behaviors across different scales of energy loss and penalty loss.

Table 3: The relative root mean squared error (rRMSE) for the total drift $\mathbf{b}$, the pseudo-potential $\psi$, and the rotational field $\mathbf{R}$, as recovered by each method.

| Method | rRMSE$_\mathbf{b}$ | rRMSE$_\psi$ | rRMSE$_\mathbf{R}$ |
|---|---|---|---|
| First-moment | 2.724e-03 | 5.339e-01 | 9.987e-01 |
| Free-energy | 7.115e-01 | 4.823e-02 | 1.005e+00 |
| Two-stage | 2.607e-03 | 2.220e-03 | 3.569e-03 |

Table 4: The relative root mean squared error (rRMSE) for the total drift $\mathbf{b}$, the pseudo-potential $\psi$, and the rotational field $\mathbf{R}$, as recovered by the proposed two-stage method with different penalties.

| Penalty | rRMSE$_\mathbf{b}$ | rRMSE$_\psi$ | rRMSE$_\mathbf{R}$ |
|---|---|---|---|
| Proposed | 2.266e-02 | 1.929e-02 | 3.495e-02 |
| $L_2$-norm | 2.176e-02 | 5.699e-01 | 9.946e-01 |

## 5 RELATED WORKS

**Strong-form based learning.** Over the last several decades, various methods have been formulated and developed for learning deterministic/sotchasitc dynamics, e.g. sparse identification of nonlinear

dynamical systems (SINDy) Brunton et al. (2016), physics-informed neural network (PINN) Raissi et al. (2019), Koopman operator theory Williams et al. (2015), to name a few. On the other hand, statistical methods including maximum likelihood methods Dietrich et al. (2023); Opper (2019); Chen et al. (2024), Gaussian processes Chen et al. (2021b); Batlle et al. (2025), kernel methods Xu et al. (2025), Wasserstein distances Ma et al. (2021), nonparametric regression techniques Lu et al. (2019; 2023); Lang & Lu (2022); Feng et al. (2024); Miller et al. (2023); Lu et al. (2022); Ding et al. (2022), etc. are introduced for learning physical laws. We remark that the existing learning frameworks are typically formulated based on the strong form of the underlying governing equations, such as (stochastic) ordinary differential equations (ODEs/SDEs) and PDEs. Different to the methods we have mentioned and some more methods in the references therein, there are some variational-form based learning frameworks are proposed aiming to enhance interpretability in learning. We refer the reader to Yu et al. (2021); Chen et al. (2024); Jin et al. (2020); Huang et al. (2024a); Chen & Tao (2021); Mattheakis et al. (2022); Bertalan et al. (2019); Finzi et al. (2020); Greydanus et al. (2019); Hu et al. (2025); Chen et al. (2020), and the references therein. This is related to the variational and weak-form based learning framework discussed below, although their loss functions are constructed based on the strong form of the differential equations.

**Variational- and weak-form based learning.** Motivated by the strong-form based learning framework, and aiming to avoid the approximation of high-order derivatives while enhancing robustness to noisy data, recent years have seen growing interest in learning frameworks based on variational and weak formulations. For example, the weak-form variant of SINDy has been proposed to learn PDEs Messenger & Bortz (2021), Hamiltonian systems Messenger et al. (2024), and mean-field equations Messenger & Bortz (2022). Similarly, numerous methods related to PINNs have been developed, such as the weak-form PINN Ryck et al. (2022), variational PINN Kharazmi et al. (2021), Physics-Informed Graph Neural Galerkin Networks Gao et al. (2022), and Weak Adversarial Networks Zang et al. (2020), to name a few. These approaches formulate the loss function based on the weak formulation of the underlying PDEs, which requires constructing a suitable class of test functions to infer the target unknown function. To avoid the explicit construction of test functions, Gao et al. (2024) proposed a self-test loss function that leverages data itself as a test function to learn Wasserstein gradient flows. Interestingly, they also observed that their loss function is closely related to the underlying energy dissipation law. In contrast to weak formulation-based approaches, several learning frameworks have been developed that are directly connected to energy structures aiming to preserve more physical structures during the learning process—whether for conservative or dissipative systems Lu et al. (2024); Lee et al. (2021); Hu et al. (2024); Zhang et al. (2024); Huang et al. (2024b); Gruber et al. (2023; 2025).

# 6 CONCLUSION

We have developed a learning framework for generalized diffusions with non-gradient structures. Focusing on the FP equation, our method learns both the pseudo-potential and rotational components by combining the first-moment evolution with the energy dissipation law. The proposed two-stage framework demonstrates strong flexibility and effectiveness in handling both gradient-driven diffusion and processes with rotational dynamics. Our approach offers several advantages: it is robust to noisy data, applicable to diffusion processes without detailed balance, and potentially extendable to high-dimensional settings. To enforce the pointwise orthogonality constraint, we introduce a weighted penalty derived via dimensional analysis. We validate our method through a series of representative numerical experiments, including ablation studies comparing the two-stage method with direct methods and evaluating the proposed weighted penalty against the standard $L_2$ penalty.

Several open problems remain for future exploration. First, applying energy laws to the learning of pseudo-potentials in general drifts without the pointwise orthogonal constraint is challenging due to the severe non-convexity of the energy-based loss. Second, extending the learning framework to the case of time-dependent pseudo-potentials via energy laws deserves future investigation. Identifying the pseudo-potential in the presence of a time-dependent drift is particularly significant, given its relevance to transformer architectures and large language models (LLMs) Bertozzi et al. (2025). However, the corresponding Fokker–Planck equation may fail to admit an energy dissipation law since the drift is time dependent, making it difficult to determine the drift and pseudo-potential at each time. Third, learning physical laws in nonlinear stochastic dynamics with nonlocal effects represents a promising direction for further research.

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

## A    COMPUTATION OF THE ENERGY-DISSIPATION LAW

In this appendix, we compute the functional derivative of the energy $\mathcal{F}[f]$ defined in (5). To begin with, we define $v \in C_c^\infty(\mathbb{R}^d)$ as an arbitrary test function. Noting that $\mathcal{F}[f + \varepsilon v]$ is continuously differentiable in $\varepsilon$, one computes the Gateaux derivative, which is

$$\frac{d}{d\varepsilon}\Big|_{\varepsilon=0} \mathcal{F}[f + \varepsilon v] = \Big\langle \frac{\delta \mathcal{F}}{\delta f}, v \Big\rangle_{L^2}$$

$$= \int_{\mathbb{R}^d} \frac{d}{d\varepsilon}\Big|_{\varepsilon=0} \Big[ (f + \varepsilon v) \ln[\tfrac{1}{2}\sigma^2(f + \varepsilon v)] + \psi(f + \varepsilon v) \Big] d\mathbf{x} = \int_{\mathbb{R}^d} v[\ln(\tfrac{1}{2}\sigma^2 f) + 1] + \psi v \, d\mathbf{x}, \quad \text{(A.1)}$$

which implies Fréchet derivative of $\mathcal{F}[f]$ is

$$\frac{\delta \mathcal{F}}{\delta f} = \ln(\tfrac{1}{2}\sigma^2 f) + 1 + \psi.$$

It follows that

$$\nabla\Big(\frac{\delta \mathcal{F}}{\delta f}\Big) = \nabla[\ln(\sigma^2 f) + \psi] = -\frac{2}{\sigma^2}\mathbf{u}, \quad \text{(A.2)}$$

where $\mathbf{u} := -[\frac{\sigma^2}{2}\nabla \ln(\sigma^2 f) + \frac{\sigma^2}{2}\nabla\psi]$. In light of the FP equation (2), one finds

$$
\begin{aligned}
\partial_t f = \nabla \cdot [(\nabla(\frac{\sigma^2}{2}f) - \mathbf{b}f)] &= \nabla \cdot [(\nabla(\frac{\sigma^2}{2}f) + \frac{\sigma^2}{2}\nabla\psi f) - \frac{1}{2}\sigma^2\mathbf{R}f] \\
&= \nabla \cdot [\frac{f\sigma^2}{2}\nabla(\ln(\sigma^2 f) + \psi) - \frac{1}{2}\sigma^2\mathbf{R}f] \\
&= -\nabla \cdot [\mathbf{u}f] - \frac{1}{2}\nabla \cdot (\sigma^2\mathbf{R}f). \quad\quad\quad (A.3)
\end{aligned}
$$

Now, we are ready to compute $\frac{d\mathcal{F}}{dt}$ with $\mathcal{F}$ defined in (5). Indeed, denoting $\langle \cdot, \cdot \rangle_{L^2}$ the $L^2$-inner product, one has from (A.2) and (A.3) that

$$
\begin{aligned}
\frac{d\mathcal{F}}{dt} &= \langle \frac{\delta\mathcal{F}}{\delta f}, f_t \rangle_{L^2} = \int_{\mathbb{R}^d} \frac{\delta\mathcal{F}}{\delta f} f_t \, d\mathbf{x} \\
&= -\int_{\mathbb{R}^d} \frac{\delta\mathcal{F}}{\delta f} \nabla \cdot [f(\sigma^2\mathbf{R} + \mathbf{u})] \, d\mathbf{x} \\
&= \int_{\mathbb{R}^d} \nabla\left(\frac{\delta\mathcal{F}}{\delta f}\right) \cdot f(\sigma^2\mathbf{R} + \mathbf{u}) \, d\mathbf{x} \\
&= -\int_{\mathbb{R}^d} \frac{2f}{\sigma^2} |\mathbf{u}|^2 \, d\mathbf{x} + \frac{1}{2}\int_{\mathbb{R}^d} [\mathbf{R} \cdot \nabla(\sigma^2 f) + \sigma^2 f\mathbf{R} \cdot \nabla\psi] \, d\mathbf{x},
\end{aligned}
$$

proving the energy evolutionary equation (6).

Moreover, by using the pointwise orthogonality condition (7) and the divergence-free condition $\nabla \cdot \mathbf{R} = 0$, since $\mathbf{R}$ is the rotation component, one finds from the integration by parts that

$$
\begin{aligned}
\frac{d\mathcal{F}}{dt} &= -\int_{\mathbb{R}^d} \frac{2f}{\sigma^2} |\mathbf{u}|^2 \, d\mathbf{x} + \frac{1}{2}\int_{\mathbb{R}^d} [\mathbf{R} \cdot \nabla(\sigma^2 f) + \sigma^2 f\mathbf{R} \cdot \nabla\psi] \, d\mathbf{x}, \\
&= -\int_{\mathbb{R}^d} \frac{2f}{\sigma^2} |\mathbf{u}|^2 \, d\mathbf{x} - \frac{1}{2}\int_{\mathbb{R}^d} [\nabla \cdot \mathbf{R}(\sigma^2 f)] \, d\mathbf{x} \\
&= -\int_{\mathbb{R}^d} \frac{2f}{\sigma^2} |\mathbf{u}|^2 \, d\mathbf{x} \leq 0,
\end{aligned}
$$

which establishes the energy-dissipation law (8).

## B  ABLATION STUDIES

### B.1  DIRECT METHODS VS TWO-STAGE METHOD

**First-moment direct method:** This method learns $\psi$ and $\mathbf{R}$ by minimizing a dynamical loss derived from the evolution of empirical first moments. Specifically, we optimize the parameters $\theta$ of the neural networks $\psi_{\text{NN}}$ and $\mathbf{R}_{\text{NN}}$ according to the following objective:

$$
\theta^*_{\psi,\mathbf{R}} = \underset{\theta}{\arg\min}\, L^{\text{dyn}}_{\psi,\mathbf{R}}(\theta), \quad\quad\quad (B.1)
$$

where $L^{\text{dyn}}_{\psi,\mathbf{R}}(\theta) = \sum_{k=1}^d L^{\text{dyn}}_{\psi,R_k}(\theta)$ and for $k = 1, \ldots, d$,

$$
L^{\text{dyn}}_{\psi,R_k}(\theta) = \sum_{j=1}^M \left\| \frac{(\mu_j)_k(t_2) - (\mu_j)_k(t_1)}{t_2 - t_1} + |\delta\mathbf{x}| \sum_{i=1}^N \frac{1}{2}\sigma^2(\mathbf{x}_i)\left[\partial_{x_k}\psi_{\text{NN}}(\mathbf{x}_i; \theta) - (R_k)_{\text{NN}}(\mathbf{x}_i; \theta)\right] f_j(\mathbf{x}_i, t_1) \right\|^2.
$$
$$(B.2)$$

**Free-energy direct method:** Alternatively, we may attempt to directly learn $\psi$ and $\mathbf{R}$ by using energy dissipation law:

$$\theta^*_{\psi,\mathbf{R}} = \underset{\theta}{\arg\min} \{L^{\text{dyn}}_{\psi}(\theta) + \lambda L^{\text{orth}}_{\psi,\mathbf{R}}(\theta)\}, \tag{B.3}$$

where $L^{\text{dyn}}_{\psi}$ is defined in (13) and

$$L^{\text{orth}}_{\psi,\mathbf{R}}(\theta) = \sum_{j=1}^{M} \left\| |\delta\mathbf{x}| \sum_{i=1}^{N} \sigma^2(\mathbf{x}_i) f_j(\mathbf{x}_i, T_1) |\nabla\psi_{\text{NN}}(\mathbf{x}_i; \theta) \cdot \mathbf{R}_{\text{NN}}(\mathbf{x}_i; \theta)| \right\|^2. \tag{B.4}$$

### B.2 Dimensional Analysis: Penalty in Energy Laws

In this appendix, we discuss our choice of penalty given by (10). The method we shall use is dimensional analysis, and we refer the reader to Drobot (1953). In detail, for any function $g$, we denote $[g]$ as the dimension of $g$. With the aid of (2), (4) and (5), we have $[\sigma^2][t] = [\mathbf{x}]^2$, $[\nabla\psi] = [\mathbf{R}] = \frac{1}{[\mathbf{x}]}$ and $[\psi] = [\ln(\frac{1}{2}\sigma^2 f)] = 1$. Moreover, one finds

$$\left[ \frac{d\mathcal{F}}{dt} \right] = \left[ \int_{\mathbb{R}^d} \frac{\partial f}{\partial t} d\mathbf{x} \right] = \left[ \int_{\mathbb{R}^d} \frac{[\sigma]^2[f]}{[\mathbf{x}]^2} d\mathbf{x} \right],$$

which implies

$$\left( \left[ \frac{d\mathcal{F}}{dt} \right] + \int_{\mathbb{R}^d} \left[ \frac{1}{2} \frac{|\nabla(\sigma^2 f)|^2}{\sigma^2 f} + \nabla(\sigma^2 f) \cdot \nabla\psi + \frac{1}{2}\sigma^2 f |\nabla\psi|^2 \right] d\mathbf{x} \right)^2 = \left( \left[ \int_{\mathbb{R}^d} \frac{[\sigma]^2[f]}{[\mathbf{x}]^2} d\mathbf{x} \right] \right)^2 = [\sigma^2]^2[f]^2[\mathbf{x}]^{2d-4}, \tag{B.5}$$

where we have used $\left[ \int_{\mathbb{R}^d} \frac{1}{[\mathbf{x}]^2} d\mathbf{x} \right] = [\mathbf{x}]^{d-2}$. Noting that the orthogonality penalty is given by (10), we have

$$\left[ \int_{\mathbb{R}^d} \sigma^2 f |\nabla\psi \cdot \mathbf{R}| \, d\mathbf{x} \right]^2 = \left[ \int_{\mathbb{R}^d} [\sigma^2][f] \frac{1}{[\mathbf{x}]^2} d\mathbf{x} \right]^2 = [\sigma^2]^2[f]^2[\mathbf{x}]^{2d-4}.$$

Otherwise, if we use the standard $L^2$ penalty, one finds the corresponding dimension is

$$\left[ \int_{\mathbb{R}^d} |\nabla\psi \cdot \mathbf{R}|^2 \, d\mathbf{x} \right] = \left[ \int_{\mathbb{R}^d} \frac{1}{[\mathbf{x}]^4} d\mathbf{x} \right] = [\mathbf{x}]^{d-4},$$

which does not match the dimension of energy dissipation rate shown in (B.5).

## C Figures and Tables

This appendix presents all the figures and some tables mentioned in Section 4.

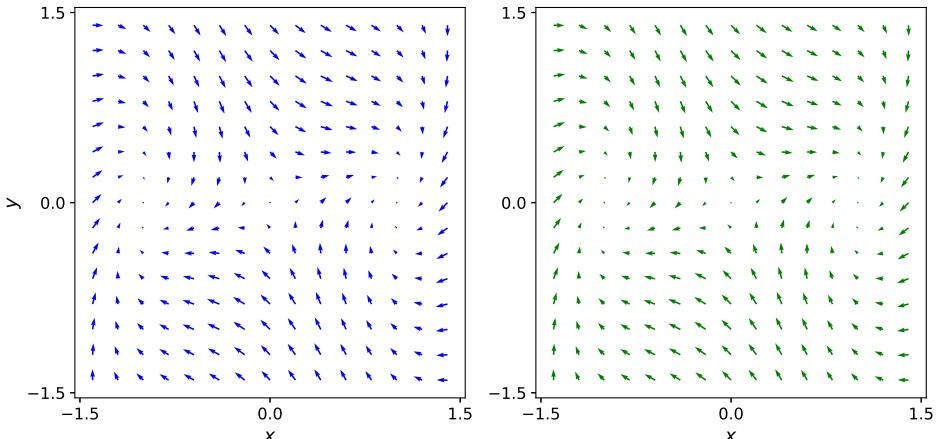

Figure 3: Comparison of the learned drift (left) $\mathbf{b}_{\text{NN}}$ with the ground truth (right) $\mathbf{b}(x,y) = \frac{1}{2(1+x^2+y^2)} \begin{bmatrix} -(x^3-x)+y \\ -y-(x^3-x) \end{bmatrix}$. The relative root mean square error is $2.266 \times 10^{-2}$.

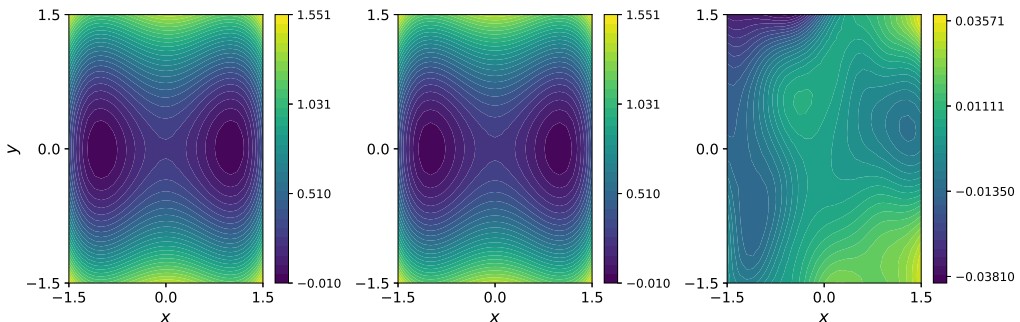

Figure 4: Comparison of the learned potential function $\psi_{NN}$ with the ground truth $\psi(x, y) = \frac{1}{4}(x^2 - 1)^2 + \frac{1}{2}y^2$. The heatmaps, shown from left to right, correspond to $\psi_{NN}$, the ground truth, and their pointwise difference. The relative root mean square error is $1.929 \times 10^{-2}$.

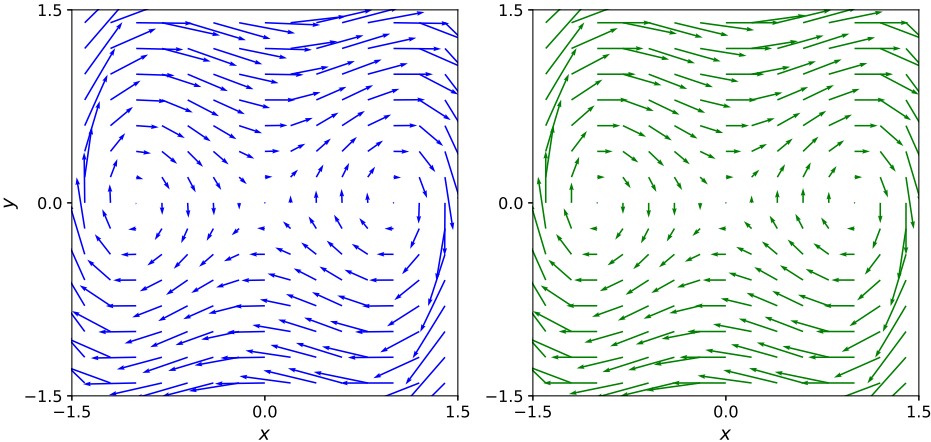

Figure 5: Comparison of the learned rotation (left) $\mathbf{R}_{NN}$ with the ground truth (right) $\mathbf{R}(x, y) = \nabla \psi^\perp = \begin{bmatrix} y \\ -(x^3 - x) \end{bmatrix}$. The relative root mean square error is $3.495 \times 10^{-2}$.

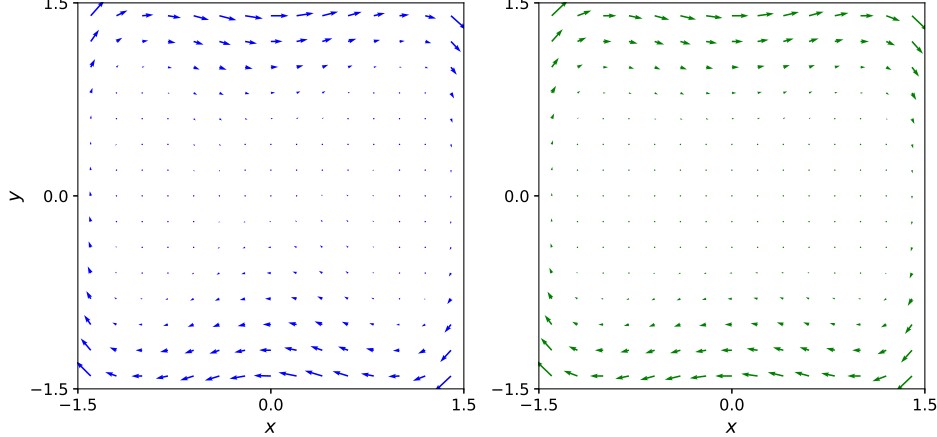

Figure 6: Comparison of the learned rotation (left) $\mathbf{R}_{NN}$ with the ground truth (right) $\mathbf{R}(x, y) = \left( \frac{1}{16}(x^2 - 1)^2 + \frac{1}{8}y^2 \right) \begin{bmatrix} y \\ -(x^3 - x) \end{bmatrix}$, using $M = 80$. The relative root mean square error is $1.692 \times 10^{-1}$.

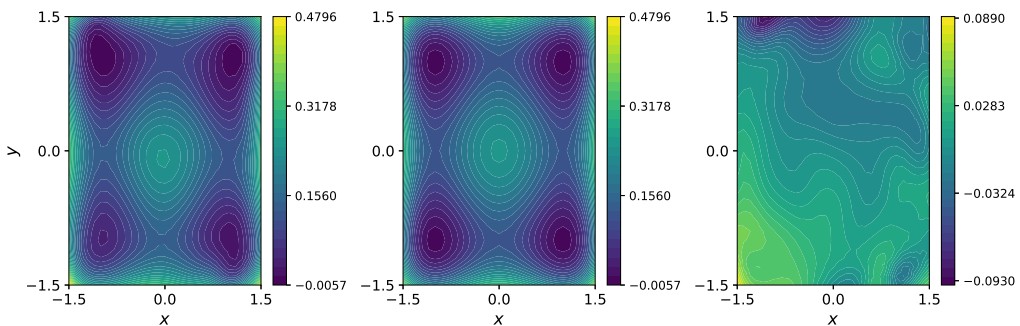

Figure 7: Comparison of the learned potential function $\psi_{\mathrm{NN}}$ with the ground truth $\psi(x, y) = \frac{1}{8}(x^2 - 1)^2 + \frac{1}{8}(y^2 - 1)^2$. The heatmaps, shown from left to right, correspond to $\psi_{\mathrm{NN}}$, the ground truth, and their pointwise difference. The relative root mean square error is $1.513 \times 10^{-1}$.

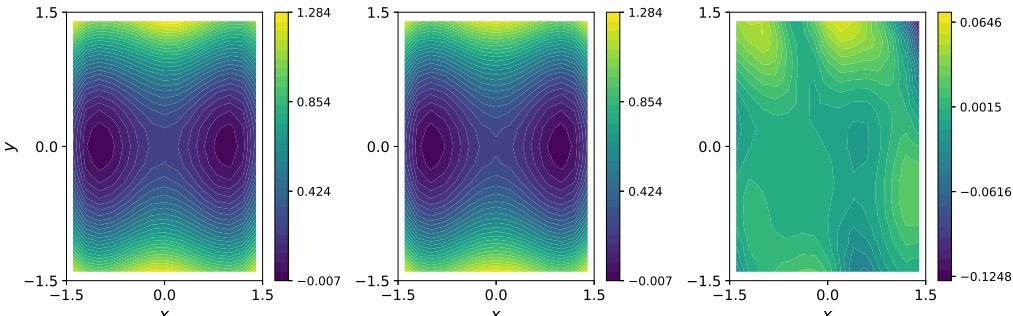

Figure 8: Comparison of the learned potential function $\psi_{\mathrm{NN}}$ with the ground truth $\psi(x, y) = \frac{1}{4}(x^2 - 1)^2 + \frac{1}{2}y^2 + \varepsilon^4 \sin(\frac{2\pi x}{\varepsilon}) \sin(\frac{2\pi y}{\varepsilon})$. The heatmaps, shown from left to right, correspond to $\psi_{\mathrm{NN}}$, the ground truth, and their pointwise difference. The relative root mean square error is $3.539 \times 10^{-2}$.

Table 5: The Impacts of $T_1$.

| $\left|\frac{d\mathcal{F}}{dt}\right|$ | $T_1$ | rRMSE$_\psi$ | rRMSE$_\mathbf{R}$ |
|---|---|---|---|
| 0.350 | 0.2 | 3.259e-01 | 5.982e-01 |
| 0.125 | 0.5 | 3.159e-01 | 5.365e-01 |
| 0.055 | 1.0 | 1.625e-02 | 2.657e-02 |
| 0.030 | 1.5 | 9.433e-03 | 2.343e-02 |
| 0.017 | 2.0 | 8.577e-03 | 2.367e-02 |
| 0.011 | 2.5 | 8.686e-03 | 2.443e-02 |
| 0.007 | 3.0 | 8.466e-03 | 2.350e-02 |

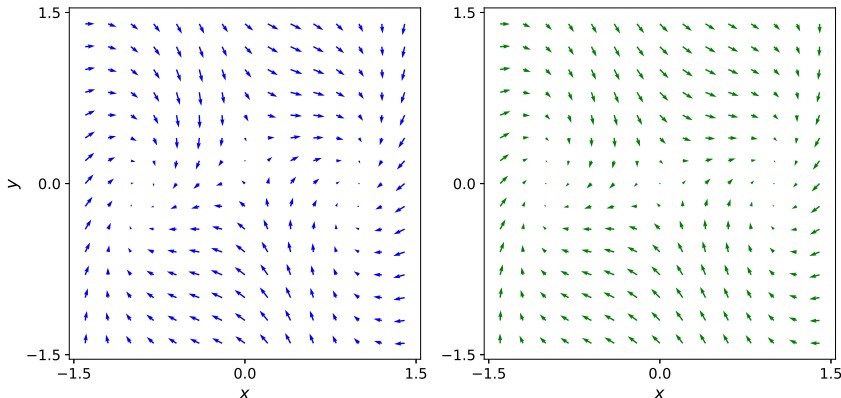

Figure 9: Comparison of the learned drift (left) $\mathbf{b}_{\mathrm{NN}}$ with the ground truth (right) $\mathbf{b}(x, y) = \frac{1}{2(1+x^2+y^2)} \begin{bmatrix} -(x^3 - x) + y \\ -y - (x^3 - x) \end{bmatrix}$. The relative root mean square error is $2.024 \times 10^{-1}$.

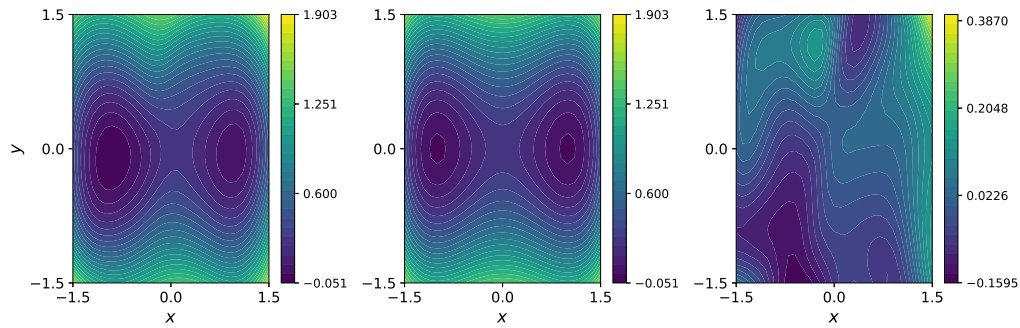

Figure 10: Comparison of the learned potential function $\psi_{\mathrm{NN}}$ with the ground truth $\psi(x, y) = \frac{1}{4}(x^2 - 1)^2 + \frac{1}{2}y^2$. The heatmaps, shown from left to right, correspond to $\psi_{\mathrm{NN}}$, the ground truth, and their pointwise difference. The relative root mean square error is $1.216 \times 10^{-1}$.

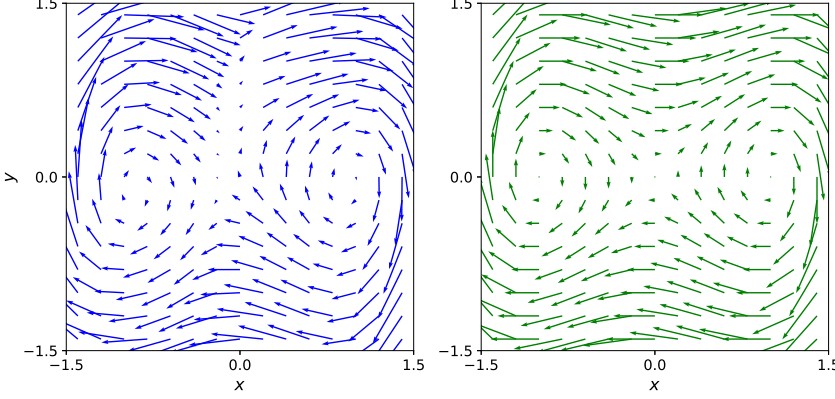

Figure 11: Comparison of the learned rotation (left) $\mathbf{R}_{\mathrm{NN}}$ with the ground truth (right) $\mathbf{R}(x, y) = \nabla \psi^\perp = \begin{bmatrix} y \\ -(x^3 - x) \end{bmatrix}$. The relative root mean square error is $3.495 \times 10^{-2}$.

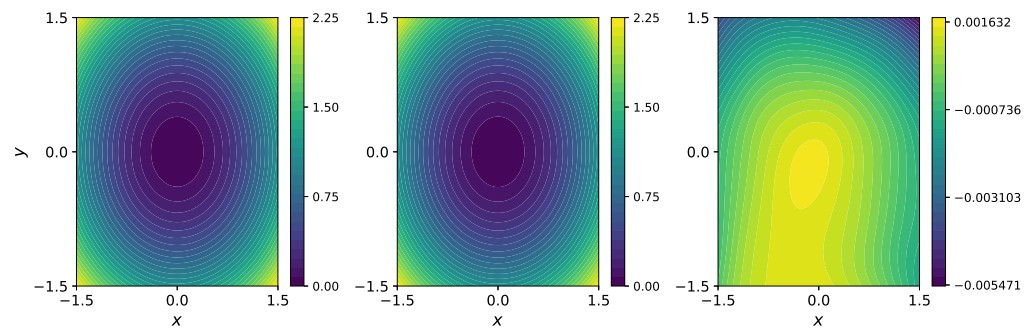

Figure 12: Comparison of the learned potential function $\psi_{\text{NN}}$ for a gradient system with the ground truth $\psi(x, y) = \frac{1}{2}x^2 + \frac{1}{2}y^2$. The heatmaps, shown from left to right, correspond to $\psi_{\text{NN}}$, the ground truth, and their pointwise difference. The relative root mean square error is $1.404 \times 10^{-3}$.

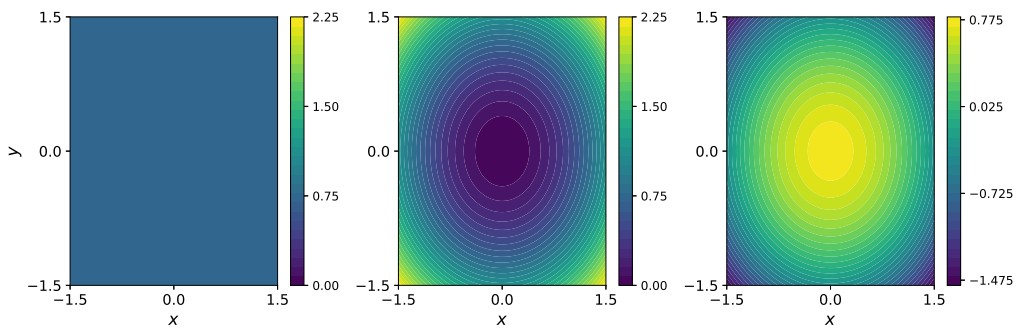

Figure 13: Comparison of the learned potential function $\psi_{\text{NN}}$ for a non-gradient system with the ground truth $\psi(x, y) = \frac{1}{2}x^2 + \frac{1}{2}y^2$. The heatmaps, shown from left to right, correspond to $\psi_{\text{NN}}$, the ground truth, and their pointwise difference. The relative root mean square error is $5.344 \times 10^{-1}$.

## D  FORMULATION OF A PDE-BASED LOSS

We minimize the loss function $L_\psi^{\text{PDE−total}}(\theta)$ below, with the aim of learn the potential for gradient and non-gradient systems.

$$L_\psi^{\text{PDE−total}}(\theta) = L_\psi^{\text{PDE}}(\theta) + \lambda L_\psi^{\text{PDE−penalty}}(\theta), \tag{D.1}$$

where

$$L_\psi^{\text{PDE}}(\theta) = \sum_{i,j=1}^{N,M} \left| \partial_t f_j(\mathbf{x}_i, t_1) + \tilde{\nabla} \cdot (b f_j(\mathbf{x}_i, t_1)) - \frac{1}{2} \tilde{\Delta}(\sigma^2 f_j(\mathbf{x}_i, t_1)) \right|^2 |\delta \mathbf{x}|$$

$$+ \sum_{i,j=1}^{N,M} \left| \partial_t f_j(\mathbf{x}_i, T_1) + \tilde{\nabla} \cdot (b f_j(\mathbf{x}_i, T_1)) - \frac{1}{2} \tilde{\Delta}(\sigma^2 f_j(\mathbf{x}_i, T_1)) \right|^2 |\delta \mathbf{x}|, \tag{D.2}$$

and

$$L_\psi^{\text{PDE−penalty}}(\theta) = \sum_{i,j=1}^{N,M} \sigma^4 f_j^2(\mathbf{x}, t_1) |\nabla \psi \cdot \mathbf{R}||\delta \mathbf{x}| + \sum_{i,j=1}^{N,M} \sigma^4 f_j^2(\mathbf{x}_i, T_1) |\nabla \psi \cdot \mathbf{R}||\delta \mathbf{x}|. \tag{D.3}$$

The penalty term $L_\psi^{\text{PDE−penalty}}$ is derived using the dimensional analysis discussed in Appendix B.2.

