# OpenReview forum: "Learning Non-Gradient Diffusion Systems via Moment-Evolution and Energetic Variational Approaches"
_ICLR.cc/2026/Conference — Submitted to ICLR 2026_

### Official Review · Reviewer_qHTP · 2025-10-28

**Soundness:** 2
**Presentation:** 3
**Contribution:** 3
**Rating:** 2
**Confidence:** 5

**Summary:**

This paper proposes a two-stage learning algorithm for generalized diffusion processes with non-gradient drift fields: first stage learning the drift fields by the first moment estimates; and second stage learning the learn the pseudo-potential parts of the drift fields by applying a physically consistent penalty in the loss to enforce orthogonality of the pseudo-potential and rotational components. The method is built on prior work such as Lu et al. (2024) and introduces a penalty term by considering the pointwise orthogonality of the pseudo-potential and rotational components to improve robustness to noisy density data. Numerical experiments in low dimensions illustrate that the method better recovers the rotational components than baseline approaches.

The paper presents an interesting and physically grounded approach to learning non-gradient diffusion drift decomposition via Helmholtz methods. However, the empirical and theoretical scope remains limited to low dimensional, synthetic settings. With stronger sensitivity analyses, realistic applications, and tighter parameter guidance, the work could become significantly more compelling.

**Strengths:**

(1) The application of Helmholtz decomposition to drift learning is conceptually compelling: separating the gradient (pseudo-potential) and divergence-free (rotational) parts aligns with physical modelling of non-equilibrium systems.
(2) The physically consistent penalty enforcing pointwise orthogonality is well motivated and matches the dimension of energy dissipation rate and may improve robustness to noisy data.
(3) The authors provide clear implementation details and present a set of representative synthetic examples, which show improved drift reconstruction over simpler baselines.

**Weaknesses:**

(1) The two-stage algorithm requires accurate density function data on a relatively large domain with dense spatial grids; this limits applicability to low-dimensional problems. The manuscript primarily uses numerical solutions of the Fokker–Planck (FP) equation as “given” density data, which raises the question: if the FP drift/diffusion terms are known (so the equation can be solved), then the learning task is less realistic.
(2 There is minimal discussion or theory guiding the choice of time windows $t_1,t_2$, $T_1,T_2$ and penalty strength $\lambda$ with respect to the underlying diffusion process (e.g., drift/diffusion regularity, relaxation time scales, spectrum of the generator).
(3) As acknowledged by the authors, the method cannot currently learn time-dependent pseudo-potentials, rotational components that vary in time, or diffusion processes with nonlocal effects. These restrictions should be discussed more clearly in terms of limitations and future work.
(4) The method depends on an accurate density field; the manuscript lacks any numerical study of how errors in the density data propagate into drift estimation and result in bias.
(5) The numerical examples remain synthetic and low-dimensional. I suggest applying the method to a more practically motivated diffusion (e.g., impurity diffusion in crystalline solids) to demonstrate relevance beyond toy problems.

**Questions:**

(1) I wonder how errors in the density data propagate into drift estimation and affect the choice of hyperparamter and result in bias in the learning objectives.
(2) Following the previous question, I wonder if some types of errors or noises in the density data are less important to the learning of rotation components .
(3) I suggest applying the method to a more practically motivated diffusion (e.g., impurity diffusion in crystalline solids) or higher dimensional examples to demonstrate relevance beyond toy problems.

---

> ### Author Response · Authors · 2025-11-26
>
> We sincerely thank the detailed comments of the reviewer.
>
> 1. We have included an example using particle data in the revised manuscript, demonstrating that our two-stage method continues to yield reliable learning results, see the lines 430-437 on pages 8-9. In addition, we investigated the effects of different temporal–spatial resolutions for collecting training data by solving the Fokker–Planck equation. As shown in Experiments 3 and 4 in Table 1, our method remains robust even when the training data have low resolution.
>
> 2. The rotation component is obtained by subtracting the learned potential component from the full drift function obtained in the first stage. In other words, once the drift function is accurately learned in the first stage and the potential component is accurately learned in the second stage, the resulting rotation component will also be reliable. Because both of our loss functions are expressed in integral form, they are naturally robust to low-resolution and noisy data, as integration can mitigate the effects of errors or noise. In addition, the energy–dissipation law is more fundamental than the corresponding Fokker–Planck equation. This structure provides an implicit regularization that guides the learning process toward physically meaningful learning results.
>
> 3. We have included examples that validate our method using particle data as well as an example in three dimensions, see the lines 417-423 on page 8. These additional results further demonstrate that our method has the potential to address more practical problems. The central objective of our paper is to develop a learning framework capable of handling non-gradient systems. Such systems have received relatively little attention within the learning community, yet they play a crucial role in non-equilibrium statistical physics and are fundamental to understanding many natural phenomena, particularly in human stem-cell regulatory networks (e.g., C. Li and J. Wang (https://journals.plos.org/ploscompbiol/article?id=10.1371/journal.pcbi.1003165), cell-fate decision dynamics (e.g., F. Chen et al. (https://pubmed.ncbi.nlm.nih.gov/37031958/), and ocean current-transprt model (e.g., §4.2 in K. Petrović et al. (https://openreview.net/forum?id=Cv84fXtQPJ&referrer=%5Bthe%20profile%20of%20Alexander%20Tong%5D(%2Fprofile%3Fid%3D~Alexander_Tong1)). We have included these references in lines 31-34 of our revised version.

---

### Official Review · Reviewer_tL3i · 2025-10-29

**Soundness:** 2
**Presentation:** 2
**Contribution:** 2
**Rating:** 2
**Confidence:** 4

**Summary:**

This paper proposes a two-stage weak-form learning framework for recovering drift decompositions in generalized diffusions without detailed balance. Stage 1 identifies the drift from first-moment evolution; Stage 2 recovers the pseudo-potential using an energy-dissipation law with a physics-motivated orthogonality penalty. The idea of combining weak-form moment evolution with energy-based learning is interesting and potentially impactful for non-gradient stochastic dynamics.

However, several parts of the theoretical formulation, numerical justification, and experimental design remain insufficiently rigorous or clearly motivated.

**Strengths:**

1. Addresses the important problem of learning non-gradient stochastic dynamics, beyond detailed-balance systems.
2. The weak-form formulation is appealing for noisy data, avoiding higher-order derivative estimation.
3. The paper provides multiple 2D diffusion examples, including noisy and rough potentials, plus ablation studies on penalty and training strategies.

**Weaknesses:**

1. Derivation of Equations (6)–(9) lacks rigor and clarity.

1.1 The transition from Eq. (4) to Eq. (6) appears ad hoc and not rigorously derived from the underlying stochastic dynamics or variational principles.

1.2 It is unclear how Eq. (8) is obtained from Eq. (6)—specifically, how the second term in Eq. (6) is eliminated and under what assumptions this simplification holds.

1.3 The statement that “we can minimize (8) in a weak form to learn the pseudo-potential and the rotation part” lacks justification. The rationale for why minimizing this functional corresponds to learning the desired decomposition should be explicitly established.

2. The constraint $\nabla\psi \cdot R = 0$ is enforced only via an integrated (global) penalty term. There is no theoretical argument showing that minimizing this global loss guarantees pointwise orthogonality. A discussion of this discrepancy and its practical implications would be important.

3. The paper provides no analysis of the consistency, bias, or variance of the proposed estimators. Without such analysis, it is unclear under what conditions the learned drift and potential converge to the true physical quantities. The good numerical results currently shown may depend strongly on the specific form of the training data rather than the generality of the method.
In particular, the dataset includes distributions at long times, which might already be close to the stationary distribution. This could artificially improve the training performance. It is recommended to quantify this effect—for example, by computing and plotting the distance between the data distribution at large t and the stationary distribution—to clarify how much of the observed accuracy stems from near-stationary data.

4. All numerical examples are synthetic 2-D toy problems. The absence of higher-dimensional or real-world cases limits the demonstration of scalability and practical relevance. Moreover, no quantitative evaluation of runtime, efficiency, or robustness across architectures is provided.

5. The assumption $b = -\tfrac{1}{2}\sigma^2\nabla\psi + \tfrac{1}{2}\sigma^2R$ is central to the method but not theoretically or physically discussed. All test cases are artificially constructed to satisfy this decomposition, which weakens the claim of general applicability. The paper should clarify under what conditions this assumption holds and how violations would affect learning performance.

6. The assumption that $\sigma$ is a scalar function is not discussed either.

**Questions:**

Please see weakness above

---

> ### Author Response · Authors · 2025-11-26
>
> We sincerely thank the reviewer for the extensive and constructive comments.
>
> In response, we have added the detailed derivations of equations (6)–(9) in Appendix A of the revised manuscript. We kindly refer the reviewer to this section for the complete mathematical steps.
> 1. On pages 3–4 of the revised version, we further clarify why the sequential minimization of the first moment and the energy law enables the recovery of both the potential component and the rotational component of the drift.
> 2. Because the penalty term contains an absolute value, its vanishing implies that ∇ψ⋅R=0 pointwise, provided the potential and rotational components are sufficiently smooth. For practical applications of the pointwise orthogonality assumption, we refer the reviewer to the work of B. Lin et al.(https://msml21.github.io/papers/id35.pdf). As demonstrated in that paper, this assumption plays an important role in large deviation theory and in the learning of quasipotentials, enabling the characterization of particle dynamics with rotational effects in the regime of sufficiently weak noise.
> 3. Regarding the consistency of our method, we have conducted additional experiments. Our results indicate that the method performs well not only when the system is near the stationary regime but also for relatively short-time dynamics. Numerical evidence supporting this claim is provided in Table 5.
> 4. We have also conducted experiments in 3D, which demonstrate that our method performs well in higher-dimensional settings. Details can be found on page 8 of the revised manuscript.
> 5. The decomposition we adopt is an assumption underlying our framework. Theoretically, the two-stage method relies on this assumption to function correctly. In practice, however, this assumption is quite mild, and we demonstrate through a variety of examples that it is widely satisfied.  This assumption fits the framework of fluctuation-dissipation law, which ensures the validity of energy dissipation relation and appears in diverse physical scenarios.
>
> 6. We provide an explanation on the lines 120-124 on page 3 of the revised manuscript. Since our primary goal is to learn the pseudo-potential, we simplify the analysis by assuming that the noise is uncorrelated. Additionally, the energy–dissipation law needs to be derived for the matrix-valued diffusion coefficient, which is beyond the scope of this paper.

---

### Official Review · Reviewer_NgmB · 2025-11-01

**Soundness:** 3
**Presentation:** 3
**Contribution:** 3
**Rating:** 4
**Confidence:** 3

**Summary:**

In the paper, the authors propose a data-driven method to learn the drift vector field of stochastic dynamic systems. Specifically, a two-stage method based on a physically consistent penalty and first-moment evolution is proposed to solve this problem.

**Strengths:**

- The investigated problem is important.
- The mathematical formula is clear written well.
- The idea of considering rotation filed and potential filed is reasonable and insteresting.

**Weaknesses:**

- The code is not provided, which limits the reproducibility of the work.

- The experimental section is a significant weakness of the paper. The most critical issue is the lack of comparisons with state-of-the-art (SOTA) baselines from top-tier conferences such as ICLR and NeurIPS. The current comparisons are limited to relatively simple methods, many of which are simplified variants proposed by the authors themselves.

- Another weakness lies in the experimental metrics, which are not intuitive, while the analysis tends to be overly subjective. For instance, in line 384, it is stated that “our method still yields reasonably reliable results.” However, it remains unclear what RMSE value qualifies as “reasonable,” as this is highly dependent on the specific scale and context of the system under study. Such claims may lead to confusion.

- The results in Figure 2 are also difficult to interpret. It is unclear from the figure whether the proposed method performs well or poorly. At the very least, a side-by-side comparison with the fields learned by SOTA methods should be provided to better illustrate the effectiveness of the proposed approach.

**Questions:**

- The experimental comparisons are currently limited to simplified baselines. Could you include comparisons with state-of-the-art methods to better demonstrate the relative performance and competitive advantage of your proposed approach?

- Please clarify the plans for releasing the source code and the experimental setup.

---

> ### Author Response · Authors · 2025-11-26
>
> We sincerely thank the reviewer for the  constructive comments.
>
> 1. Our work considers a novel scenario in which the underlying dynamics are stochastic with a non-gradient drift. To address the resulting analytical challenges, we propose a two-stage framework specifically designed for this setting. To the best of our knowledge, such a framework for handling stochastic dynamics with non-gradient drift has not been developed in the existing literature.  Despite this, we have included an example on a PDE-based method, which is widely used in science and engineering. The learning results show that the PDE-based method can successfully recover the potential for gradient systems from data, but fails to do so for non-gradient systems.
>
> 2. We shall post our code on github once our paper is accepted.

---

### Official Review · Reviewer_f8na · 2025-11-05

**Soundness:** 4
**Presentation:** 4
**Contribution:** 2
**Rating:** 6
**Confidence:** 3

**Summary:**

The authors propose a two-stage method for learning the drift of SDEs in the setting where the ground truth SDE does not satisfy a detailed balance condition. The approach is based on decomposing the drift into a gradient (pseudo-potential) and a rotational term. The framework uses snapshots of the probability density function generated from different initial Gaussian densities, captured at both short and intermediate times. Stage 1 learns the total drift via first-moment evolution, and Stage 2 learns the decomposition using an energy dissipation law.

**Strengths:**

- The authors identify a tractable subset of the difficult non-gradient SDE learning problem: systems where the drift decomposition satisfies a pointwise orthogonality constraint.
- The paper proposes a novel two-stage learning framework that cleverly combines moment-evolution and energy-dissipation principles.
- The numerical evaluation, while limited to 2D examples, is thorough. It effectively demonstrates the method's robustness to significant data noise, rough potentials, and non-canonical rotations.
- The work is very well presented.

**Weaknesses:**

- The method's primary weakness is its data requirement. It assumes access to full, gridded snapshots of the density function, which is unrealistic in most practical applications where data typically consists of sparse, noisy particle trajectories.
- The reliance on gridded data and Riemann sums for integration raises concerns about the method's scalability to high-dimensional problems due to the curse of dimensionality.
- The authors claim applicability to biology and engineering, but the experiments are limited to 2D toy problems.
- The entire method is contingent on the pointwise orthogonality constraint. There is limited discussion on the prevalence of this assumption in real-world systems or how the method's performance degrades if this constraint is only approximately satisfied.

**Questions:**

- The noise robustness experiment is a good inclusion. However, could the authors provide a more formal analysis of error propagation? Specifically, if the density f were estimated from sparse data (e.g., via KDE), how would that estimation error propagate through the loss functions?
- How does this method perform in practical applied settings?

---

> ### Author Response · Authors · 2025-11-26
>
> We sincerely thank the reviewer for the insightful comments.
>
> 1. The particle-to-density approach for the EnVarA loss has been discussed in Lu et al. (https://arxiv.org/pdf/2412.04480), which demonstrates that the learning framework remains reliable even when the data are sparse, as probability densities can be estimated from particle data via KDE. To further validate this setting within our two-stage method, we have added an example in the revised manuscript, see the lines 430-437 on pages 8-9. The numerical results indicate that our method continues to perform reliably, particularly in capturing key dynamical features such as the wells of the target potential function. It is worth emphasizing that, in many real-world applications, the primary goal is to recover the dominant dynamics of the underlying system rather than an exact potential function. In practice, recovering an exact potential function is impossible due to noisy, sparse, or even partially observed data.
>
> 2. As mentioned in the point above, we have included an example that validates our method using particle data. In addition, we have provided a three-dimensional example, see the lines 417-423 on page 8. Both examples further demonstrate that our method has the potential to address more practical problems. Indeed, the primary goal of our paper is to propose a learning framework capable of handling non-gradient systems—an area that has received relatively little attention in the learning community, despite its importance in non-equilibrium statistical physics and its role in modeling many natural phenomena, particularly in human stem-cell regulatory networks (e.g., C. Li and J. Wang (https://journals.plos.org/ploscompbiol/article?id=10.1371/journal.pcbi.1003165), cell-fate decision dynamics (e.g., F. Chen et al. (https://pubmed.ncbi.nlm.nih.gov/37031958/), and ocean current-transprt model (e.g., §4.2 in K. Petrović et al. (https://openreview.net/forum?id=Cv84fXtQPJ&referrer=%5Bthe%20profile%20of%20Alexander%20Tong%5D(%2Fprofile%3Fid%3D~Alexander_Tong1)). We have included these references in lines 31-34 of our revised version.

---

### Meta-Review · Area_Chair_jKUS · 2026-01-06

**Summary:**

This paper proposes a method for estimating an SDE from observational data, in which the SDE includes a drift term with terms other than the gradient. The proposed method first estimates the drift term and then decomposes it into a gradient and a rotation component, and the reviewers acknowledge the novelty of this idea. However, the reviewers' primary concerns are related to the numerical experiments, and I believe there is significant room for improvement in the paper. So I cannot recommend accepting this paper.

**Reviewer Concerns:**

The main concerns are the requirement for the density function $f$ as observed data and the lack of sufficient comparison with other methods. Regarding the first concern, while the authors conducted additional experiments, the revised paper does not appear to provide sufficiently detailed explanations. Therefore, this concern does not seem fully addressed. Regarding the second concern, the authors explain that few comparable methods exist due to the novelty of the problem setting. However, I believe that this is primarily due to the first concern: the problem setting where the density function is provided as data. Consequently, the concerns are considered unaddressed.

**Reviewer Scores:**

The scores were 6-4-2-2. Unfortunately, no discussion between the authors and the reviewers is available. However, for the reasons above, I do not believe that the scores have changed significantly.

---

### Decision · Program_Chairs · 2026-01-26

Reject